# Safety Messaging Boosts Parental Vaccination Intention for Children Ages 5–11

**DOI:** 10.3390/vaccines10081205

**Published:** 2022-07-28

**Authors:** Zhihan Cui, Lu Liu, Dan Li, Sherry Jueyu Wu, Xinyue Zhai

**Affiliations:** 1UCLA Anderson School of Management, Los Angeles, CA 90095, USA; sherry.wu@anderson.ucla.edu; 2Department of Economics, New York University Shanghai, Shanghai 200122, China; ll4748@nyu.edu; 3Yale School of Medicine & Yale School of Public Health, New Haven, CT 06520, USA; dan.li@yale.edu; 4Law School, University of Pennsylvania, Philadelphia, PA 19104, USA; xinyzhai@pennlaw.upenn.edu

**Keywords:** COVID-19, vaccination, messaging, children, parents, safety

## Abstract

The COVID-19 vaccination rate among children ages 5–11 is low in the U.S., with parental vaccine hesitancy being the primary cause. Current work suggests that safety and side effect concerns are the primary reasons for such vaccine hesitancy. This study explores whether this hesitancy can be mitigated with information interventions. Based on theories of health decision making and persuasion, we designed four information interventions with varying contents and lengths. We wrote two messages on vaccine safety (a detailed safety-long message and a succinct safety-short message), explaining the vaccine’s lower dosage, low rate of side effects, and the rigorous approval process. We also had two messages on protection effects (protect-family, protect-child). We combined these four messages with a vaccine-irrelevant control message and compared their effects on parental vaccine intention. We measured the parental vaccination intention using a 0–6 Likert scale question. Among the four intervention groups, we found that the short version of the safety message increased the average vaccination intention by over 1 point compared to the control arm, while the other three interventions failed to show significance. Specifically, these effects are particularly pronounced (around 2 points) for Republican parents who had a much lower initial intention to vaccinate their children. Our study highlights the importance of concise and to-the-point information rendering in promoting public health activities and therefore has important policy implications for raising vaccination intentions among parents, especially those leaning towards more conservative political affiliation.

## 1. Introduction

COVID-19 has been posing increasing risks to the health and quality of life of children. COVID-19 has become one of the leading causes of death in children between the ages of 5 and 14 in the United States over the past few years [1]. Additionally, long COVID may impact children even if they had only mild symptoms during infection [2]. Furthermore, children’s infection may also lead to more extensive spread of COVID-19 to other high-risk age cohorts [3]. To cope with this dual threat, vaccination is among the most effective means. Solid clinical research has demonstrated the safety, immunogenicity, and efficacy in children of the BNT162b2 COVID-19 vaccine [4], and the favorable benefit–risk ratio is also justified in real-world settings in the U.S. and Italy [5,6,7].

Despite these well-documented foundations, the COVID-19 vaccination rate of children ages 5–11 is still strikingly low in the U.S. As of 6 July 2022, the full vaccination rate in the United States had reached 76.9% for all adults and 91.6% for people over 65; meanwhile, although the vaccine had been available for eight months for children of ages 5–11, this rate was still only about 30%, which was only half the rate of 12–17 [8]. Specifically, Republican parents’ reluctance is formidable, with less than 10% having fully vaccinated their children. Since the children’s vaccination decisions are solely dependent on their parents, parental hesitancy toward pediatric vaccination has posed significant public health risks as parents are decision-makers for children’s vaccination [9,10].

Many researchers have explored these causes empirically. Safety (including side effect) concerns significantly contribute to parental COVID-19 vaccine hesitancy, even though the safety of the vaccines for children 5–11 has already been effectively ensured. Both a nationwide survey by Kaiser Family Foundation (KFF) [11] and prior studies [12,13,14,15,16,17] highlighted that worries about safety and side effects are the dominant hesitancy concerns of U.S. parents, and this pattern is also robust in other countries [18,19]. In most studies, more than 50% of parents reported concerns that COVID-19 vaccines would incur severe side effects [12,13,14,15,16]. Moreover, many studies documented that the need for more information was also playing a role [13,16,17,18]. The doubts on the benefit of pediatric vaccines are smaller, with some worrying about the effectiveness [17,18] and even fewer declaring that COVID-19 is not a dangerous disease to prevent [13].

These empirical findings imply that, if we want to improve the vaccination rate, effective persuasion is urgently needed to reduce parental COVID-19 vaccine hesitancy for children [20,21]. Proper narratives and message contents are crucial for effective persuasion and avoiding backfiring [20,22]. Studies of social psychology and health decision making have both presented multiple theories and models, such as the Health Belief Model (HBM), the Protective Motivation Theory (PMT), and the “Deficit Model”, which can provide direct insights for our study.

The HBM [23,24] is one of the most well-known models for health decisions and can be used in pediatric vaccination topics [25]. It defines the key factors that one should highlight to maximize health behavior influences—perceived susceptibility, perceived severity, perceived benefits, perceived barriers, cue to action, and self-efficacy. In the case of increasing vaccination intention by messaging interventions, HBM identifies the need to highlight perceived benefits of the vaccines. Apart from the HBM, the PMT [26,27] predicts preventive health behaviors by analyzing people’s responses to triggers that assess the potential threats. Both imply that vaccine messages should emphasize the ability of vaccines to save lives as well as the safety of vaccines [28]. Additionally, because COVID-19 vaccines are a novel medical intervention and the general public is even more unacquainted with the pediatric version, especially in the year 2021, researchers [29,30] argue that the “deficit model” (lack of information on risks and benefits as a major determinant of hesitancy) may also be a good perspective despite the recent criticism of this model (the criticism claims that this model ignores other important psychological factors that shape the risk/benefit perception) [31]. This model implies the necessity to offer information that is likely to be previously unknown. The convergence of the theoretical and empirical findings has set up solid foundations for our information contents.

Besides contents, the way interventions are presented is also crucial according to psychological insights. Effective persuasion requires careful design of materials based on social psychological models. The most widely accepted theories in discussing the methods of persuasion are the dual-process models, such as the Elaboration Likelihood Model (ELM) [32,33,34] and the Heuristic Systematic Model (HSM) [35,36]. Both models have been applied to health decisions, sometimes together with the health belief and motivation models [33,34,36]. The strategies are somewhat opposite for these two systems. For persuading through the central system, or the more “rational” pathway, longer materials with accurate argumentations would perform better, while for persuading through the peripheral system, or the more “heuristic” pathway, more succinct and catchy messages would be more effective [37].

Recent studies, with few talking about the case of children ages 5–11, mainly focused on using information interventions for adult COVID-19 vaccination, and they generated mixed results [29,38,39,40,41]. Ashworth et al. [38] (emphasizing benefits and safety), Motta et al. [39] (emphasizing collective benefits), and Palm et al. [40] (emphasizing efficacy and safety) all found significant effects of persuasion on vaccine intention. However, Kerr et al. [29] and Loomba et al. [41] detected null effects of information interventions. It seems that the effectiveness of messages may be determined by multiple conditions, further justifying the importance of a careful and tactful design to graft this idea to pediatric COVID-19 vaccination.

Built upon the theoretical and empirical foundations above, our paper aims to investigate information interventions to mitigate vaccine hesitancy. An online survey experiment was designed to explore the effects of four different information interventions with varying content and length on parental vaccination intentions. Our paper begins by laying out the design of the information interventions. In our paper, “intervention”, “group”, and “message” are used interchangeably. We then analyze the effects of the interventions on parental vaccination intentions and discuss the applications and policy implications of our findings in depth. Our findings will provide valuable insights into communication strategies to boost parental vaccine intention. The study can provide guidance to politicians, health officials, and educators seeking to increase pediatric vaccination rates.

## 2. Materials and Methods

### 2.1. Research Design

We designed information interventions and tested their effects on changing the subjects’ intention to vaccinate their children (parental vaccination intention). In Section 2.1, we include text designs and randomization.

#### 2.1.1. Intervention Texts

In the introduction, we discussed both theoretical and empirical support to justify our messaging strategy of persuasion: using proper content in a well-organized way to reduce parents’ safety concerns and thereby increase parental vaccination intention.

First, we carefully designed “what to say”. Empirical evidence implies that safety concerns are primary, so we should design messages containing information that mitigate the safety concerns. Specifically, the deficit model suggests that lack of information may cause a higher risk of hesitancy, so our information interventions should try to involve new information not frequently mentioned but that may be crucial for decisions, such as the lower dosage (1/3 of the adult/teenage dosage). Then, both HBM and PMT discuss the importance of the benefits of pediatric vaccination. Thus, as an important comparison, we should design some interventions on the benefit side.

Second, we focused on “how to say it”. The literature still leaves room for us to conduct exploratory research on this question, so we compared two messaging strategies here: persuasion through the peripheral system, which may be more suitable for our study based on the deficit model, suggested a succinct and catchy way, while persuasion through the central system suggested the use of detailed messages with accurate argumentations.

Based on the aforementioned foundations, we first designed two interventions on important facts about vaccine safety for children ages 5–11. We provided information that (1) the dosage is lower, and so is the side effect rate; (2) the development and authorization processes are rigorous with three-stage trials all guaranteed; and (3) parents have the right to transparent vaccine information. The *Safety-short* intervention presented short, concise information about these issues, and the *Safety-long* intervention presented long, detailed information.

Then, on the benefits side, we designed two information interventions that directly highlighted the benefits of vaccinating their child with other factors unmentioned. The protect-child intervention presents information about how COVID-19 vaccines could prevent the children from adverse outcomes of contracting the virus, and the protect-family intervention presents information on how COVID-19 vaccines could protect the whole family by preventing the child from infections, both adapted from Ashworth et al. [38]. Participants in the control condition were presented with no information on vaccination, but instead, a vaccine-irrelevant message about working ethics. The content of each text is briefly summarized in Table 1; the full text of each condition can be found in the Appendix A (See Appendix A, Appendix A).

#### 2.1.2. Randomization

We designed a survey experiment on Amazon Mechanical Turk (MTurk) to examine the effects of four information interventions on parental vaccination intention. Participants were randomized via the Qualtrics randomization function to one of the five groups and then asked about their attitudes towards vaccination (details in Section 2.3 and Section 2.4). Subjects were blinded to the randomization.

### 2.2. Participants and Sample Size

The inclusion criteria were: (1) adults aged ≥ 18 years, (2) having at least one unvaccinated child between ages 5–11. For data quality concerns, we further excluded incomplete responses or responses that failed the one-item attention check entrenched in the vaccine hesitancy scale.

If there was only one child in the family, we took the response variable directly. When a participant had two unvaccinated children, we treated them as two separate observations because what ultimately matters is the vaccination rate of all children. Among more than 1900 respondents from MTurk, we obtained *n* = 243 valid observations of parents, generating a total children pool of *N* = 324.

### 2.3. Data Collection

Participants were recruited via CloudResearch on MTurk, with requirements of CloudResearch approval and high approval rates (>80%). After informed consent, study participants reported demographic information and were thereby screened by the criteria in Section 2.2. Then, each of the eligible participants was randomly exposed to a message (four intervention groups and one vaccine-irrelevant control group) and was required to read the material for at least 1 min. Afterwards, participants were asked about their ratings of vaccination intention for children and their general perceptions of this issue with a children-oriented COVID vaccination hesitancy scale. The study was approved by the Institutional Review Board at UCLA (IRB#21-002085) and preregistered in the AsPredicted platform.

### 2.4. Outcome Variables

#### 2.4.1. Primary Outcome Variable 1: Parental Vaccination Intention

Our primary outcome variable, parental vaccination intention, was measured through a 0–6 Likert scale question. For parents with only one eligible child, we asked, “Do you plan to have your child take a COVID-19 vaccination shot within the coming 30 days?”; and for parents with two or more eligible children, we followed the strategy of Ashworth et al. [38] and asked the same question about their eldest and youngest child.

#### 2.4.2. Primary Outcome Variable 2: Children-Oriented COVID-19 Vaccine Hesitancy Scale

We developed a 9-item questionnaire derived from the vaccine hesitancy scale from Shapiro et al. [42] by adopting its language to fit the scenario of COVID-19 vaccination for children. The descriptive statistics and correlations for the scale are shown in Table 2. The items and the correlation matrix of the questionnaire are shown below. The Cronbach’s alpha of this scale is 0.931.

#### 2.4.3. Important Covariates

We also asked about important demographics, including education level, household size, household income, age, and gender. Summary statistics are shown in Section 3.1.

Another important covariate is political affiliation. COVID-19 vaccination is highly politicized, so partisanship may impact our results. We measured this through a question directly asking party identity, from “strong Republican” to “strong Democrat” on a 1–7 scale. We assigned subjects reporting “strong Republican”, “Republican”, and “independent leaning Republican” to the Republican subsample, and vice versa. Subjects who reported “Independent” belonged to neither subsample.

We also included questions about the participants’ personal histories and experiences of vaccination and their children’s infection histories and recent COVID-like symptom experiences. The full description of the variables and experiment is available in the Appendix A.

### 2.5. Statistical Analysis

We relied on ordered Probit regressions and linear regressions (to estimate the average effect size) with and without control variables. Standard errors were clustered at the family (respondent) level for our main regressions. All analysis was conducted by Stata 14. We included the balance statistic results in Appendix A of the Appendix A, and the process of factor analysis and other regression specifications in the Supplementary Analysis.

## 3. Results

### 3.1. Background Characteristics of the Participants

Within the 243 eligible participants, about half of the participants were between 18 and 34 years old (47.7%). About one third of the participants were male (32.1%). A total of 54% of the participants were fully vaccinated, and slightly more than 50% have experienced some side effects. Among them, ¼ had experienced severe side effects. The division between Democrat and Republican subsamples was quite even (each about 40%), and 19.3% claimed to be fully Independent. Within the 324 children included in the 243 families, 23% had been infected with COVID-19 in the past. A detailed report of the summary statistics is shown in Table 3.

### 3.2. Parental Vaccination Intention

The average vaccination intention of participants was scored on a 0–6 Likert scale (Figure 1). In the control group, the average score is 1.68, indicating that the parents assigned in this group have a low intention to vaccinate their children, similar to the KFF findings [7]. The group with the short safety message (Safety-short) generated the highest parental vaccination intention (average score = 2.70), which statistically significantly differs from the control group (ordered Probit, *p* = 0.01**; linear, *p* = 0.02**). This effect size is highly comparable to the most effective private benefit message in Ashworth et al. [7]. However, average scores in other groups fail to exhibit statistically significant effects (*Protect-child*: average score=1.88; *Protect-family*: average score = 1.57; *Safety-long*: average score=1.84; all *p* > 0.2). These findings robustly indicate that the only effective option is a short, concise message on vaccine safety, consistent with the literature that safety concerns are the primary source of pediatric COVID-19 vaccine hesitancy. Other interventions, which may respectively suffer from different deficiencies (protection problems may be a primary concern, and long messages may cause attention fatigue), do not appear effective in mitigating parental vaccine hesitancy. These findings are robust to observation setups or the addition of simple controls (family size, age of child, partisanship, etc.).

### 3.3. Mediation Analysis on Vaccine Hesitancy

To better characterize the mechanisms of the interventions, we modified the well-renowned vaccine hesitancy scale [14] to a version that explicitly focused on COVID-19 vaccines for children (see Table 2). We did the same regressions as above with the average score for this scale and found that the only group with a statistically significant contribution was also the Safety-short intervention, generating a 0.54-point (*p* = 0.009) increase compared to the control group out of a 5-point Likert scale. This finding is highly consistent with the vaccination intention result above.

Moreover, we used a mediation analysis to test the mechanisms of our intervention. According to the Protective Motive Theory (PMT), the scores of the vaccine hesitancy scale should play the mediating role between the intervention and the final vaccination intention. Our exploratory mediation analysis with the medeff package in Stata [43] showed that the vaccine hesitancy scale mediated the interventions and the vaccination intention, with 98% of the total effect mediated. This is consistent with theoretical prediction, further justifying our internal validity. Detailed results of the mediation analysis is in Appendix A of the Appendix A.

### 3.4. Partisanship Heterogeneity Effects

Given that COVID-19 vaccination behaviors are typically politicized [44,45], and Republican parents are much more reluctant to vaccinate their children [7], it is especially meaningful to look at the asymmetric effects of information interventions on members of different parties. We detected a significant cross-partisan discrepancy: the Safety-short message was tremendously effective in the Republican subsample (Control: 1.10; Safety-short: 3.13; *p* < 0.01), while it was insignificant in the Democratic subsample (Control: 2.60; Safety-short: 2.87; *p* > 0.2). This suggests that our message performed better with Republican parents. This result is robustly justified in the interaction analysis (See Appendix A, Appendix A). The Safety-short arm drove the vaccination intention of Republicans three times as much as those in the control group, even slightly surpassing their Democratic counterparts (the difference is not statistically significant).

## 4. Discussion

This paper managed to use information interventions to deal with the pediatric vaccination hesitancy among parents of children ages 5–11 in the U.S. We discovered that a brief and catchy message on vaccine safety might help promote parental vaccination intention, but other messages might not, and that the effect was suggestively larger for Republican parents.

Our study has three important implications. It supplements the current literature on parental vaccine hesitancy and vaccine messaging by offering an effective intervention that managed to reduce it and tested it through randomized controlled trials. We found that the safety-short message had a statistically significant effect that is robust to most model specifications, indicating a good internal validity. Our study also acts as a causal test of the theoretical models and past empirical work [12,13,14,15,16,17,18] that detected robust patterns that safety concerns contribute to parental vaccine hesitancy. We suggest that the parental vaccine hesitancy may not be extremely sticky, and at least something can be done to reduce it.

Second, in accordance with the rules of thumb on debunking [22] and persuasion [32,33,34,35,36], we found that concise and to-the-point messaging is fundamental for effective boosting. One can improve their messaging effectiveness by briefly summarizing the points that parents are most concerned about and highlighting them in a concise paragraph. We point out that it would be helpful for institutions to establish a carefully designed notice within one paragraph to summarize that vaccines are safe and why. We suggest that institutions may find it effective to involve more procedure-related information on pediatric vaccine safety, such as the lower dosage and rigorous approval processes, instead of only emphasizing the conclusion that the vaccines are safe or documenting complicated supporting scientific evidence.

Third, despite their low pediatric vaccination rate of about 10% [7], Republican parents are still open to vaccination persuasion. When exposed to appropriate messaging of vaccines, their vaccination intention could reach a level similar to Democrats. On the contrary, the vaccination intention among Democrats may have already touched the ceiling: Democrats are generally more pro-vaccine, but for those Democrats who have finally decided to refuse vaccination of children long after the vaccine has been available, it is significantly harder to change their minds. This partisanship heterogeneity perfectly confirms the theory of planned behavior [46,47]: information interventions may perform efficiently only in those groups whose minds have not been saturated by existing information. Democrats may have already made up their minds after gaining all relevant information in their local communities. These findings have significant policy implications, suggesting that we should improve the information rendering efficiency in vaccine messaging, and that there may still be potential to encourage Republican parents to vaccinate their children, despite the traditional view that they are very unlikely to do so.

Some potential limitations of this study include:(1)The mechanisms of all the interesting heterogeneity effects are not fully explored, and the explanations are therefore not thorough. The limited size (mainly due to difficulties in data collection: we could recruit only 15–20 effective participants per day according to the basic screening rules of CloudResearch, as the proportion of participants with unvaccinated children ages 5–11 is low) and sample representativeness may also lead to our inability to consummately address the mechanism problems. For instance, although the partisanship difference is intriguing, we need to acknowledge that this heterogeneity effect needs to be further addressed by more statistical evidence in larger samples. Further work is needed to identify the detailed mechanisms of the attitude change process.(2)The external validity of these information interventions is still pending study. Vaccination intention does not necessarily equate to vaccination behavior. Despite the high correlation between intention and behavior [48], there are still gaps between them [49], and the long-term effects of information interventions still await future study. It will be interesting and meaningful for future researchers to test these interventions in the field to assess the real-world power of vaccine messaging.

## 5. Conclusions

This paper tested the effectiveness of four information interventions on the parental vaccination intention of their children at ages 5–11. We found that a brief and catchy message that resolved the safety concerns successfully improved the parental vaccination intention by 1 point on a 0–6 scale compared to the control group, and there was supportive evidence that such an increase was more significant for Republican parents. On the contrary, messages that focused on protection or a longer and more detailed message that focused on the same content failed to generate significance. Our finding also supports the idea that the effect of safety concerns on vaccination intention is causal. The most important policy implications for governments and health workers are to establish concise, to-the-point, and easy-to-remember messages about safety in their communication materials.

## Figures and Tables

**Figure 1 vaccines-10-01205-f001:**
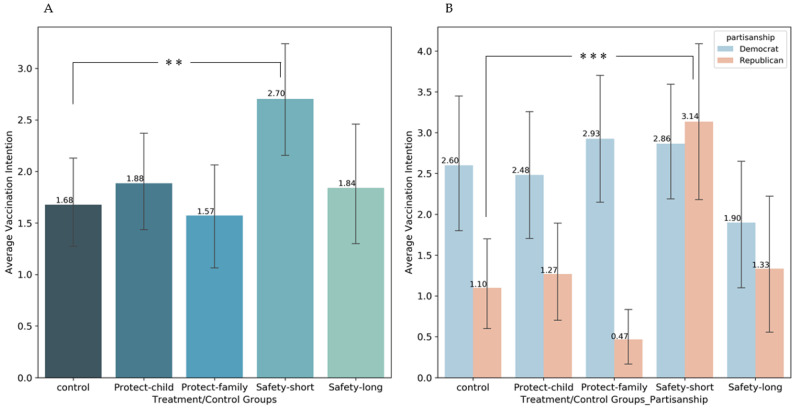
Average vaccination intention of parents in different groups. (**A**). Average vaccination intention of parents in different groups by whole sample; (**B**). Average vaccination intention of parents in different groups by partisanship. Error bars reflect ±1 SEM. ** *p* < 0.05, *** *p* < 0.01.

**Table 1 vaccines-10-01205-t001:** Overview of message content in experimental conditions.

Condition	*n*	Content	Reference	Length in Words
Control	47	A placeholder, no extra information on vaccination.	-	112
Protect-child	55	Information about how COVID-19 vaccines could prevent the children from adverse outcomes of contracting the virus.	Ashworth et al. (2021) [38]	137
Protect-family	44	Information about how COVID-19 vaccines could protect the whole family by preventing the child from infection.	Ashworth et al. (2021) [38]	130
Safety-short	57	Short, concise information about the lower doses (and thus, a lower side effect rate) of pediatric COVID-19 vaccines and the rigorous processes of approval.	U.S. FDA’s webpage	152
Safety-long	40	Long, detailed information about the lower doses (and thus, a lower side effect rate) of pediatric COVID-19 vaccines and the rigorous processes of approval.	U.S. FDA’s webpage	359

**Table 2 vaccines-10-01205-t002:** Means, standard deviations, and correlations for children-oriented COVID-19 vaccine hesitancy scale.

Title 1	Mean	SD	1	2	3	4	5	6	7	8	9
1. COVID-19 vaccines are important for my child’s health.	2.74	1.42									
2. Getting a COVID-19 vaccine is a good way to protect my child/children from the disease.	2.81	1.40	0.90 *								
3. COVID-19 vaccines are effective for children.	2.88	1.30	0.81 *	0.83 *							
4. Having my child vaccinated is important for the health of others in my community.	2.91	1.44	0.85 *	0.85 *	0.80 *						
5. Children’s COVID-19 vaccines offered by the government program in my community are beneficial.	2.84	1.38	0.86 *	0.91 *	0.85 *	0.84 *					
6. The information I receive about COVID-19 vaccines from the vaccine program is reliable and trustworthy.	2.93	1.33	0.76 *	0.78 *	0.78 *	0.76 *	0.79 *				
7. Generally I do what my doctor or health care provider recommends about COVID-19 vaccines for my child/children.	3.23	1.27	0.51 *	0.53 *	0.54 *	0.47 *	0.55 *	0.48 *			
8. COVID-19 vaccines carry more risks than influenza vaccines.	3.49	1.21	−0.29 *	−0.32 *	−0.26 *	−0.27 *	−0.30 *	−0.26 *	−0.05		
9. I am concerned about serious adverse effects of children’s COVID-19 vaccines.	3.94	1.21	−0.30 *	−0.33 *	−0.29 *	−0.27 *	−0.28 *	−0.29 *	−0.06	0.65 *	

⁎ *p* < 0.001.

**Table 3 vaccines-10-01205-t003:** Background characteristics of the participants (*n* = 243).

Characteristics		*N*	%
Sociodemographic characteristics			
Gender			
	Male	78	32.1
	Female	163	67.08
	Others	2	0.82
Age group, years			
	18–34	116	47.74
	35–54	118	48.56
	55 or above	9	3.7
Partisanship identity			
	Strong Republican	28	11.52
	Republican	46	18.93
	Independent Leaning Republican	27	11.11
	Independent	47	19.34
	Independent Leaning Democrat	33	13.58
	Democrat	49	20.16
	Strong Democrat	13	5.35
Ethnicity			
	White	179	73.66
	Hispanic	15	6.17
	Black	19	7.82
	Asian	21	8.64
	Others	9	3.7
Highest level of education			
	Some high school or less	0	0
	High school graduate	35	14.4
	Completed some college, but no degree	42	17.28
	Associate’s degree	31	12.76
	Bachelor’s degree	102	41.98
	Master’s or professional degree	27	11.11
	Doctorate degree	6	2.47
Number of people in the household			
	1	1	0.41
	2	29	11.93
	3	68	27.98
	4	94	38.68
	5	37	15.23
	6 or more	14	5.76
Annual household income			
	$25,000 or less	26	10.7
	$25,000–$35,000	25	10.29
	$35,000–$45,000	29	11.93
	$45,000–$55,000	29	11.93
	$55,000–$70,000	52	21.4
	$70,000–$85,000	23	9.47
	$85,000–$100,000	15	6.17
	$100,000–$120,000	13	5.35
	$120,000–$140,000	12	4.94
	$140,000 or more	19	7.82
History of COVID-19 and COVID-19 vaccination			
Own status of COVID-19 vaccination			
	I am fully vaccinated and had got a booster (3rd shot).	69	28.4
	I am fully vaccinated but had not gotten a booster.	68	27.98
	I am partially vaccinated (with one shot).	9	3.7
	I am not vaccinated.	97	39.92
Ever experienced any side effects or not			
	Yes	68	46.58
	No	78	53.42
Severity of the vaccination side effect experiences			
	Very severe	9	13.24
	Severe	8	11.76
	Moderate	25	36.76
	Mild	22	32.35
	Very mild	4	5.88
Child ever tested positive for COVID-19			
	No	124	82.67
	Yes	26	17.33
	Youngest child (if more than one child in the family)		
	No	66	70.97
	Yes	27	29.03
	Oldest child (if more than one child in the family)		
	No	70	75.27
	Yes	23	24.73

## Data Availability

The data will be available upon reasonable request to the corresponding authors.

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
