# Peer review of "Safety Messaging Boosts Parental Vaccination Intention for Children Ages 5–11"

_vaccines, 2022, doi:10.3390/vaccines10081205_

Round 1
Reviewer 1 Report
The manuscript entitled ''Safety Messaging Boosts Parental Vaccination Intention for 2 Children Ages 5-11' highlights the importance of concise and to-the-point information rendering in promoting public health activities and therefore has important policy implications for raising vaccination intentions among parents, especially those leaning towards more conservative political affiliation.
Although the paper has big interest for the readers but the discussion should be go deeper and the message to the community should be more clear.
Author Response
Dear Vaccines Editor and Reviewer 1,
We sincerely thank the editor and reviewers for your constructive and insightful comments and appreciate the opportunity to provide revisions. We are grateful for reviewers’ comments highlighting the policy implications of our paper for raising vaccination intentions among parents, especially those with more conservative political affiliations.
We have included all editor and reviewer comments and responded to them individually, indicating exactly how we addressed each concern or problem and describing the changes we have made. All revisions are marked using the “Track Changes” function. The revisions have been approved by all authors. We sincerely thank you for your time.
Reviewer 1 Comments
Comment :
“Discussion should go deeper and the message to the community should be more clear.”
Author Response:
Thank you so much for recognizing the importance of the topic.
We added two paragraphs which provide deeper insights into the practical implications of this study, which now reads:
“Our study has three important implications. It supplements the current literature on parental vaccine hesitancy and vaccine messaging by offering an effective intervention that managed to reduce it and tested it through random controlled trials. We find that the safety-short message had a statistically significant effect that is robust to most model specifications, indicating a good internal validity. Our study also acts as a causal test of the theoretical models and past empirical work [12-18] that detected robust patterns that safety concerns contribute to parental vaccine hesitancy. We suggest that the parental vaccine hesitancy may not be extremely sticky, and at least something can be done to reduce it.
Second, in accordance with the rules of thumb on debunking [22] and persuasion [32-36], we found that concise and to-the-point messaging is fundamental for effective boosting. One can improve their messaging effectiveness by briefly summarizing the points that parents are most concerned about and highlighting them in a concise para-graph. We point out that it would be helpful for institutions to establish a carefully de-signed notice within one paragraph to summarize that vaccines are safe and why. We suggest that institutions might find it effective to involve more procedure-related in-formation on pediatric vaccine safety, such as the lower dosage and rigorous approval processes, instead of only emphasizing the conclusion that the vaccines are safe or documenting complicated supporting scientific evidence.”
Reviewer 2 Report
The COVID-19 pandemic has had significant humanitarian and economic impacts on many countries. The authors touch upon a socially important problem - "vaccination hesitancy", and suggest ways to increase the motivation of parents to vaccinate their children in the United States.
The authors analyzed several messages capable of raising the motivation of parents to vaccinate. It turned out that short information about the safety and absence of side effects significantly increases the motivation of parents to vaccinate their children. Moreover, the greatest effect was achieved among groups with conservative political affiliation, which initially had a reduced motivation to vaccinate their children.
Main comments
The abstract is not clear enough.
The authors inform the reader in the abstract: “generated two messages for each topic, one concise and one detailed message”. And after authors say: “For the safety message,…. “it is not clear if safety message is “concise” or “detailed” probably both are right.
Further, authors suddenly introduce “Among the four treatment arms…” without explanation what this “treatment” mean. However in manuscript text we find 5 groups of information messages “Safety-short”, “Safety-long”, “Protect-child”, “Protect-family”, and “Control messages”. The authors need to simplify text and clarify the main experimental groups and present their results more clearly in the abstract.
A number of expressions are used in the abstract and in the text of the article, which, in our opinion, are misleading about the type of research. For example “two information treatments” “Safety-short treatment”. The word "treatment" in the abstract further confuses the reader, who perhaps, will read only the abstract without delving into the details of the article. The purpose of the study was not to cure, but only to convey information to make the right decision about vaccination.
We understand, indeed, that the literature uses the expression “treatment” in relation to the applied useful information to improve the healthy behavior of respondents, but still this “slang” is not used immediately in the text of the abstract without introducing the reader to the terminology of the study.
In addition, it would be great to have information about the full text of the messages that were sent to the respondents in the supplementary data. The reader will benefit from evaluating information about the "safety" and "protective effect of vaccination" to learn facts that might convince conservative respondents to vaccinate their children. It seems that the article will only benefit from this. Otherwise, the manuscript contains only well-known principles for presenting information that have long been used in advertising for the sale of consumer goods.
It would also be great to clarify the design of the experiment. For example, it is not clear whether families with more than one child and in which one of the children has already been vaccinated were evenly distributed among groups. Perhaps this significantly influenced the results of the study. Families where at least one of the two children has been vaccinated are more likely to have other children vaccinated.
Minor comments
Line 45-53 Please simplify the sentence: “We present the important facts about the safety of vaccines for children ages 5-11 in order to directly respond to parents’ concerns, and those facts include a specially designed low dosage, the complete and rigorous development and authorization processes with three-stage trials all effectively guaranteed, and parents’ right to transparent and clear information about the vaccine.”
Line 105-106 “In the control group, the average score is 1.68, indicating that the children assigned in this group have a low intention to be vaccinated,…” – It is not clear from the sentence who makes the decision children or their parents?
Please provide a definition “a place-holding message”.
Author Response
Dear Reviewer 2,
Thanks so much for all your suggestions!
Letter to Reviewer 2 Comments
Comment :
“The authors inform the reader in the abstract: ‘generated two messages for each topic, one concise and one detailed message.’ And after the authors say: ‘For the safety message,…. ‘it is not clear if safety message is ‘concise’ or ‘detailed.’”
Author Response:
Thanks so much for this helpful comment. We made modifications in the abstract to avoid this confusion. Now, the corresponding sentence in the abstract is as follows, “We wrote two messages on vaccine safety (a detailed safety-long message and a succinct safety-short message), both explaining the vaccine’s lower dosage, the low rate of side effects, and the rigorous approval process.”
Comment :
“The authors suddenly introduce “Among the four treatment arms…” without explanation what this “treatment” means. However, in manuscript text we find 5 groups of information messages: “Safety-short”, “Safety-long”, “Protect-child”, “Protect-family”, and “Control messages”. The authors need to simplify text and clarify the main experimental groups and present their results more clearly in the abstract.”
Author Response:
Thank you so much for this helpful comment. We replaced “treatment” with “intervention” throughout the paper to avoid confusion with the general meaning of the word. We agree with reviewer 2 that "intervention" is a more appropriate term for describing the function of the messages.
Furthermore, to make a distinction between the four information interventions and the one vaccine-irrelevant control message, we have revised the text, which now reads:
“We test whether this hesitancy can be mitigated with information interventions. Based on theories of health decision and persuasion, we designed four information interventions which vary in their contents and lengths. We wrote two messages on vaccine safety (a detailed safety-long message and a succinct safety-short message), explaining the vaccine’s lower dosage, the low rate of side effects, and the rigorous approval process. We also had two messages on protection (protect-family, protect-child). We combined these four messages with a vaccine-irrelevant control message and tested their effects on parental vaccine intention.”
Comment :
“The word "treatment" in the abstract confuses the reader, who perhaps, will read only the abstract without delving into the details of the article. This expression is misleading in that it might give the false impression about the purpose of this article. It should not be used immediately in the text of the abstract without introducing the reader to the terminology of the study.”
Author Response:
Thank you very much for pointing this out. For clarification, we replaced “treatment” with “intervention” and “message” for the entirety of this article. We have also defined the terminology and explained keywords in the introduction.
Comment :
“It would be great to have information about the full text of the messages that were sent to the respondents in the supplementary data. The reader will benefit from evaluating information about the "safety" and "protective effect of vaccination" to learn facts that might convince conservative respondents to vaccinate their children.”
Author Response:
Thank you so much for this insightful comment. In the SI Appendix, we have included the full text of the message and the survey questionnaire. The main article includes an introduction to the contents of the messages as well as a link that leads directly to the messages.
Comment :
Line 45-53 Please simplify the sentence: “
We present the important facts about the safety of vaccines for children ages 5-11 in order to directly respond to parents’ concerns, and those facts include a specially designed low dosage, the complete and rigorous development and authorization processes with three-stage trials all effectively guaranteed, and parents’ right to transparent and clear information about the vaccine.
Author Response:
- Thank you so much for pointing out this potential confusion! The corresponding sentence (now in line XX-YY) reads like this: “We informed that (1) the dosage is lower, and so is the side effect rate; (2) the development and authorization processes are rigorous with three-stage trials all guaranteedï¼› and (3) parents have the rights to get transparent vaccine information.” We believe that the current version is easier to comprehend.
Comment :
Line 105-106 “In the control group, the average score is 1.68, indicating that the children assigned in this group have a low intention to be vaccinated,…” – It is not clear from the sentence who makes the decision, children or their parents?
Author Response:
Thank you very much for pointing this out. Now it is clarified as: “indicating that the parents assigned to this group have a low intention to vaccinate their children”, since parents are the ones who make the decision.
Comment :
Please provide a definition “a place-holding message”.
Author Response:
Thank you very much for pointing this out. We agree that this wording may cause confusion. Thus, we changed it to “vaccine-irrelevant” to show that the control message had nothing to do with vaccines.
Reviewer 3 Report
The differential effectiveness of a short versus long message in changing prior intentions has been the subject of analysis by various theoretical models of attitude change and persuasion. Therefore, a review of these is required, and otherwise the reader may get the wrong idea from their work.
Among such models, the expectancy-value models (e.g., the health belief model, the theory of reasoned action and the theory of protective motivation) stand out, which share the idea that intention depends on two elements: the subjective probability that it will lead to the expected results (behavior change), and the value attributed to these.
Likewise, the issue has been addressed by the response elaboration model, which suggests that intention change will be possible if it is motivating for the individual and he/she has the capacity or availability to carry it out.
Depending on these conditions, the cognitive processing of the message will be unique. Thus, messages contrary to what, we think, will be processed peripherally and those that ratify our previous ideas will be processed according to a central route.
The peripheral route consists of processing information based on easily acceptable general principles (e.g., "people who seem to know what they are doing are trustworthy"), or by appealing to emotions, such as fear (e.g., "the consequences of not getting vaccinated are more harmful than those of getting vaccinated"). The central route relies on argumentation, the more exhaustive the better because it validates what I think.
Either of these models provides a more complete explanation of the results obtained than the length of the message, and they provide a more understandable framework for the differential reaction of Republicans and Democrats. In this regard, I would also point out that they do not provide information on the procedure followed to establish this classification.
In conclusion, I believe that their work, as you acknowledge, requires more significant and extensive theoretical grounding.
On the other hand, unless I am mistaken, your work presents methodological problems that cannot be overlooked. Among these, the non-random assignment of participants to the control and treatment groups stands out, as this makes it impossible to ensure that the groups are comparable so that the results are unreliable.
The structure of the study also does not conform to the standard that governs in these cases; for example, information on the study design is presented in the introduction, and the discussion is brief and incorporates variables that have not been measured or, at least, we do not know how they have been measured (e.g., classification into Republicans and Democrats).
In summary, the study has too many limitations to recommend its publication, and there are so many that, even if it were redone, it would be very difficult to avoid biases that would affect its internal and external validity.
Author Response
Dear Reviewer 3,
Thank you so much for offer these helpful comments!
Reviewer 3 Comments
Comment :
Provide a review of theoretical models of attitude change and persuasion.
Author Response:
Thank you so much for this insightful comment. In the revised version, we added a review of theoretical models on attitude change and persuasion, including the Health Belief Model (HBM), the Protective Motivation Theory (PMT) and the “Deficit Model”. The review reads as follows:
“These empirical findings imply that, if we want to improve the vaccination rate, effective persuasion is urgently needed to reduce parental COVID-19 vaccine hesitancy for chil-dren [20-21]. Proper narratives and message contents are crucial for effective persuasion and avoid backfiring [20, 22]. Studies of social psychology and health decision making have both presented multiple theories and models, such as the Health Belief Model (HBM), the Protective Motivation Theory (PMT), and the “Deficit Model”, which can provide us direct insights for our study.However, there is still limited exploration on this topic. Recent works about messaging and hesitancy mainly focus on adults vaccinating themselves [7], while few talks about parental vaccination hesitancy towards their chil-dren ages 5-11.
The HBM [23-24] is one of the most well-known models for health decisions and can be used in pediatric vaccination topics [25]. It defines the key factors that one should highlight to maximize health behavior influences – perceived susceptibility, perceived severity, perceived benefits, perceived barriers, cue to action, and self-efficacy. In the case of increasing vaccination intention by messaging intervention, HBM identifies the need to highlight perceived benefits of the vaccines. Apart from the HBM, the PMT [26-27] predicts preventive health behaviors by analyzing people’s responses to triggers that assess the potential threat. Both imply that vaccine messages should emphasize the ability of vaccines to save lives as well as the safety of vaccines [28]. Additionally, because COVID-19 vaccines are a novel medical intervention and the pediatric version is even a more unacquainted issue to the general public, especially in the year 2021, researchers [29-30] argue that the “deficit model” (lack of information on risks and benefits as a major determinant of hesitancy) may also be a good perspective despite the recent criticism on this model [31]. This model implies the necessity to offer information that is likely to be previously unknown. The convergence of the theoretical and empirical findings has set up solid foundations for our information contents.
Besides content, the way of presenting interventions is also crucial according to psychological insights. Effective persuasion requires careful design of materials based on social psychological models. The most widely accepted theory in discussing the way of persuasion is the dual-process models, such as the Elaboration Likelihood Model (ELM) [32-34] and the Heuristic Systematic Model (HSM) [35-36]. Both models have been applied to health decisions, sometimes together with the health belief and motivation models [33-34, 36]. The strategies are somewhat opposite for these two systems. For persuading through the central system, or the more “rational” pathway, longer materials with ac-curate argumentations would perform well, while for persuading through the peripheral system, or the more “heuristic” pathway, more succinct and catchy messages, would be more effective [37].
Recent studies, with few talking about the case of children ages 5-11, mainly focus on using information interventions for adult COVID-19 vaccination, and they generated mixed results [29, 38-41]. Ashworth et al. [38] (emphasizing benefits and safety), Motta et al. [39] (emphasizing collective benefits), and Palm et al. [40] (emphasizing efficacy and safety) all found some significant effects of persuasion on vaccine intention. However, Kerr et al. [29] and Loomba et al. [41] detected null effects of information interventions. It seems that whether messages would be effective may be determined by multiple conditions, further justifying the importance of a careful and tactful design to graft this idea to pe-diatric COVID-19 vaccination.
”
Comment :
“The structure of the study also does not conform to the standard that governs in these cases; for example, information on the study design is presented in the introduction, and the discussion is brief and incorporates variables that have not been measured or, at least, we do not know how they have been measured (e.g., classification into Republicans and Democrats).”
Author Response:
Thank you so much for clarification on the issues of the structure. In the current version, we rearranged the major contents of the paper to make it in accordance with the standardized structure of the Vaccines journal.
In addition, we defined clearly how the classification process is done in “2.4.3 Important Covariates”. We assigned subjects reporting “strong Republican”, “Republican”, and “independent leaning Republican” to the Republican subsample, and vice versa.
Comment :
Concerns regarding randomization.
Author Response:
We appreciate this comment. We made modifications to clarify the confusion. We make clarifications in the paper that assignment to the five groups (one control group and four intervention groups) are completely random.
Comment :
Concerns regarding internal and external validity.
Author Response:
We greatly appreciate this insightful comment. To justify our internal validity, in the revised version we stressed that we used a mediation analysis, demonstrating that vaccine hesitancy scale mediates the intervention and the vaccination intention. The added paragraph reads:
“Also, we use a mediation analysis to test the mechanisms of our intervention. Ac-cording to the Protective Motive Theory (PMT), the scores of the vaccine hesitancy scale should play the mediating role between the intervention and the final vaccination in-tention. Our exploratory mediation analysis with the medeff package in Stata [43] shows that the vaccine hesitancy scale mediates the interventions and the vaccination intention, with 98% of the total effect mediated. This is consistent with theoretical prediction, further justifying our internal validity.”
Also, we pointed out the robustness of our data in the discussion of the implication of this study, further proving our internal validity, which reads:
“It supplements the current literature on parental vaccine hesitancy and vaccine messaging by offering an effective intervention that managed to reduce it and tested it through random controlled trials. We find that the safety-short message had a statistically significant effect that is robust to most model specifications, indicating a good internal va-lidity.”
Finally, in the last parts of the Discussion section, we discuss the limitations of externality validity and talk about future perspectives.
“(2) The external validity of these information interventions is still pending study. Vaccination intention does not necessarily equal to vaccination behavior. Despite the high correlation between intention and behavior [48], there are still gaps between them [49] and the long-term effects of information interventions still await future study. It will be interesting and meaningful for future researchers to test these interventions in the field to assess the real-world power of vaccine messaging.”
Round 2
Reviewer 1 Report
The authors addressed the requested changes.
Reviewer 2 Report
Thank you for the significant revision of the manuscript.
Please don't forget to download the suplimentary materials
I wish you good luck in publishing your work.
Best regards
Reviewer 3 Report
I thank you for the attention you have paid to my comments, I sincerely did not expect you to be able to make all the necessary corrections in such a short period of time. In my opinion, the study is now more comprehensible and adjusted to the standards of the journal, and the contribution they make to the problem addressed can be better discerned.